# Successful Prevention of Antimicrobial Resistance in Animals—A Retrospective Country Case Study of Sweden

**DOI:** 10.3390/antibiotics10020129

**Published:** 2021-01-29

**Authors:** Martin Wierup, Helene Wahlström, Björn Bengtsson

**Affiliations:** 1Department of Biomedical Sciences and Veterinary Public Health, Swedish University of Agricultural Sciences (SLU), 750 07 Uppsala, Sweden; 2National Veterinary Institute (SVA), 751 89 Uppsala, Sweden; helen.wahlstrom@gmail.com (H.W.); bjorn.bengtsson@sva.se (B.B.)

**Keywords:** veterinary medicine, food animal production, antimicrobial resistance, antimicrobial use, disease prevention, disease eradication, antimicrobial growth promoters, organised health control, policies and guidelines

## Abstract

The misuse and overuse of antibiotics have resulted in an alarmingly high prevalence of antimicrobial resistance (AMR) in human and animal bacteria. European monitoring programmes show that AMR occurrence in food animals is lower in Sweden than in most other EU Member States and that the use of antibiotics for animals is among the lowest in Europe. In this retrospective country case study, we analysed published documents to identify factors contributing to this favourable situation. A fundamental factor identified was early insight into and sustained awareness of the risks of AMR and the need for the prudent use of antibiotics. Early and continuous access to data on antibiotic use and AMR made it possible to focus activities on areas of concern. Another factor identified was the long-term control and eradication of infectious animal diseases, including coordinated activities against endemic diseases, which reduced the need to use antibiotics. Structures and strategies for that purpose established at the national level have since proven useful in counteracting AMR as an integral part of disease prevention and control, guided by a “prevention is better than cure” approach. A third factor identified was consensus among stakeholders on the need to address AMR and their cooperation in the design and implementation of measures.

## 1. Introduction

The misuse and overuse of antibiotics have accelerated the development and emergence of antibiotic resistance (AMR) and resulted in an alarmingly high prevalence of AMR in human and animal bacteria [1]. To avert a crisis, in 2015, the World Health Organization (WHO) adopted a Global Action Plan on AMR to reduce the incidence of infections through disease prevention measures and to optimise the use of antibiotics in humans and animals [1]. Resistant bacteria circulating in animal populations threaten both animal and human health, so multi-sector collaboration between relevant sectors is required, as reflected by the tripartite collaboration on AMR agreed in the Global Action Plan by the Food and Agriculture Organisation (FAO), World Organisation for Animal Health (OIE) and WHO [2].

Recognition of the risks associated with the use of antibiotics in food animals, in particular, the use of antibiotics for growth promotion (AGP), began to emerge in the 1960s [3]. Some 30 years later, the global focus turned to AMR in food animals [4]. In 1997, the WHO recommended banning the use of AGPs [5] and formulated global principles for the containment of AMR, including the monitoring of AMR and antibiotic usage [6,7]. Shortly thereafter, the European Union (EU) decided to phase out the use of AGPs by 2006 [8] and initiated the monitoring of AMR in bacteria from animals in 2003 [9] and of antibiotic sales for animals in 2010 [10]. These monitoring programmes have shown that the use of antibiotics in animals is lower in Sweden than in other EU Member States [11] and that the occurrence of AMR in Sweden is among the lowest in Europe [12].

As postulated in the EU One Health Action Plan against AMR [13], lessons learnt from successful strategies in individual Member States could be valuable for other countries and support the objectives of the WHO action plan [1]; the aim of this paper was to identify those key factors that have contributed to the current favourable situation in Sweden.

## 2. Food Animal Production

Over recent decades, animal production has undergone a dramatic change in terms of mechanization, automation and management along with improvements of the animal genetic capacity and feed efficiency. This has resulted in a decrease in the number of farms producing food animals along with an increase in herd size, as summarized here for the three major production sectors in Sweden.

In dairy production, the number of farms in Sweden producing food animals has decreased, while the herd size on the remaining farms has increased. For example, the number of dairy herds has decreased by around 99% during the past 70 years, from approximately 307,000 in 1951 to 3477 in 2018 (Figure 1).

During the same period, the number of dairy cattle decreased by about 80% (1,564,000 in 1951 to 319,387 in 2018), but due to a higher yield per cow, Swedish milk production has only decreased by about 31% [14]. During the past 40 years, the average milk yield per cow per year has increased from about 5000 to 8900 L [15], and dairy production in Sweden is currently among the top performers in Europe [16]. The average herd size increased from about 14.1 cows per herd in 1979 to 91.9 in 2018 (Figure 2).

The most recent change towards larger dairy herds has occurred in herds with >199 cows, which are increasing in number, whereas the number of herds with fewer than 99 cows is decreasing. (Figure 3).

Total pig production in Sweden increased from about 90,000 tons of meat in 1945 to 330,000 tons during the 1980s–1990s, and then decreased to the current (2019) 250,000 tons, with about 2.5 million pigs slaughtered annually [14]. The number of pig farms decreased from 350,000 in 1925 to 1300 in 2018 [14]. Key data on production efficiency in 2019 in controlled Swedish herds show an average daily weight gain for fattening pigs of 948 g, with 27.1 piglets produced per sow and year [17].

The largest change in the food animal sector has been in poultry production, where the level today is eight times higher than in the 1950s [18]. Commercial broiler production increased from 74,000 tons in 1995 to 158,000 tons in 2019 [14], and in 2018, a total of 98.5 million broiler chickens were produced [19]. As seen for milk and pig production, the size of poultry farms has increased and, for commercial broiler production, the number of farms has also increased [19]. 

### Summary of Key Factors

Food animal production in Sweden has undergone structural changes, with the concentration of production in larger units and high productivity from an international perspective.

## 3. Prevention and Control of Infectious Diseases

### 3.1. Diseases Controlled by National Authorities

#### 3.1.1. Major Epizootic Diseases

During the 18th–20th centuries, Sweden frequently experienced outbreaks of major epizootic diseases, which in general, were effectively controlled by national authorities [20]. In addition to the major outbreaks listed in Table 1 and outbreaks of rinderpest, contagious bovine pleuropneumonia, rabies and anthrax, Swedish authorities were early in legally defining other disease outbreaks as epizootic and accordingly controlled them in the same powerful way, aiming for disease freedom. These diseases included bovine genital campylobacteriosis, fowl typhoid and pullorum disease (*Salmonella* Gallinarum and *S*. Pullorum) [20].

The eradication of bovine tuberculosis (bTB) and bovine brucellosis required substantial long-term efforts. Both diseases were probably introduced by early imports of infected breeding animals during the middle of the 19th century or earlier and became widespread. In 1937, 30% of the cattle slaughtered in southern Sweden had macroscopic lesions of tuberculosis [23], while in 1944, 16,000 (6%) of the cattle herds in Sweden were infected with bovine brucellosis [24]. During the period 1897–1933, bTB was controlled by a low-intensity and mainly unsuccessful control programme based on tuberculin tests, but in 1934, the control was extended to a national eradication programme, and Sweden was declared free from bTB in 1958 [25]. Efforts to eradicate bovine brucellosis were started in 1944, based on nationwide monitoring using the Abortus Bang Ring Test, and by 1962, the infection was eradicated [24]. As part of these eradication programmes, a nationwide network of veterinary laboratories was built, which facilitated the subsequent control of other diseases (see below).

#### 3.1.2. Salmonella

The early initiation of *Salmonella* control by national authorities in Sweden more than 60 years ago is unique from an international perspective because zero tolerance for *Salmonella* in the whole feed-to-food chain was applied from the start. The control of *Salmonella* is therefore described in more detail here.

*Animals*: In 1953, *S. Typhimurium* spread from a domestic slaughterhouse in Sweden and caused the death of 90 people and more than 9000 cases of illness [26]. In another outbreak, 500 people were infected by meat imported from South America [27]. These outbreaks of salmonellosis in humans highlighted the need to control *Salmonella*, which was enforced by Swedish regulations in 1961 [28]. Thereby, findings of *Salmonella* from animals, humans, feed or food were made notifiable to the authorities. Infected herds are subject to restrictions until the infection has been eradicated, but the use of antibiotics to clear farms of infection is not permitted [21,29].

In poultry production, the control of *Salmonella* is stricter, and flocks infected with any serovar of *Salmonella* are destroyed. In broiler production, a voluntary control programme run by the authorities, including the pre-slaughter testing of flocks, was initiated in 1970 [30]. The programme became mandatory in 1984 [31]. In order to avoid possible *Salmonella* contamination in broiler abattoirs, in 1991, the broiler industry introduced the pre-slaughter testing of laying hens [20]. This was later included in a voluntary control system in layer production, in response to the pandemic spread of *S.* Enteritidis in the late 1980s [32] and became mandatory in 1994 [33]. Currently, all commercial poultry production is under control for *Salmonella* [21,29].

*Feed*: The control of *Salmonella* in feed was initiated as a voluntary programme run by the industry in 1958, when the assessment of outbreaks of anthrax from imported meat and bone meal also frequently revealed simultaneous contamination with *Salmonella* [34]. The programme was later extended to the bacteriological testing of other risk ingredients such as soy and rape meal, heat treatment of poultry feed (1972) and testing of all non-heat-treated feed (1987), and in 1991, HACCP-based control of the whole production chain was implemented [20]. The voluntary control became compulsory in 1993 [35], and under current legislation, crushing plants and feed mills must be closed for decontamination upon any detection of *Salmonella* on the “clean side” [36]. The control of *Salmonella* in feed production, including imported feed ingredients, most certainly prevented the introduction of this pathogen into animal production [37]. This control, and that on imported breeding animals (see below), is considered to have prevented the introduction of *S.* Enteritidis into Swedish poultry production during its pandemic spread in the late 1980s [32].

*Food*: Food of animal origin contaminated by any serovar of *Salmonella* is considered unfit for human consumption, and in 1971, this was formalised in national legislation [38]. This zero tolerance of any contamination by *Salmonella*, in particular, of carcasses after slaughter, is considered to have contributed to compliance with the pre-harvest control of *Salmonella*.

### 3.2. Diseases Controlled by the Industry

The introduction of more intensive and specialized food animal production processes increased health disorders caused by endemic diseases, mainly respiratory infections. Trade in animals, especially to specialist and large-scale production units, was identified as a risk factor. The pig industry, therefore, established a health advisory veterinary organisation to prevent these infections as early as 1945, and provided health counselling services (here referred to as health control) as a complement to the conventional treatment of sick animals mainly performed by officially employed district veterinary officers [39]. The laboratory facilities established in all regions of Sweden for the eradication of bTB and bovine brucellosis also had sufficient capacity for providing diagnostic services for health control. A basic task for health control in pig production was to certify herds permitted to sell live animals on the market, with the mandate to withdraw certification in the case of health problems. In 1954, the Swedish dairy sector established a corresponding health control for bovine mastitis [20], and health controls were later introduced in the beef and sheep sectors [39]. It is estimated that today, close to 100% of all commercial dairy, pig and poultry producers in Sweden participate in such health controls.

#### 3.2.1. Organised Health Controls

Following an inquiry by the national veterinary services in the late 1960s [40], a parliamentary decision in 1969 clarified that the industry should be financially responsible for health controls. Prior to that decision, health controls, including the laboratory diagnostic support described above, were subsidised by the state. However, an act on the control of animal diseases established the concept of organised health control. The aim was to control specific diseases and health issues of national importance, and organised health controls were therefore eligible for state financial support. The concept was specified as follows: “*The state, through a trusting collaboration with the industry, can make the control work so that it also benefits the state*” [40]. The organised health control system initiated is managed by the industry and led by a control organisation, usually the veterinary organisation affiliated with the cooperative slaughterhouses or dairy associations. The organised health controls follow rules approved by the Swedish Board of Agriculture, the supervising authority. According to the current legislation [41], the aim of this voluntary control is to “*stop or prevent the spread of infections and diseases and to improve the health of farm animals*”. All producers can join and must be treated equally by the control organisation, for example, regardless of whether they are linked to a cooperative or private slaughterhouse, which commonly competes in the red meat sector. To achieve joint action by affiliated farmers, each organised health control has a steering committee representing all key stakeholders and the supervising authority. This structure facilitates compliance with the control and the final eradication of a disease, for example, by preventing individual producers from delivering animals to slaughter or milk to dairies if they refuse to follow the rules or join the control [42].

The organised health controls have proven to be an efficient tool for preventing and controlling diseases. A major focus was to limit risks related to the trade of animals. Examples of infections for which organised health controls were implemented for control or eradication are given in Table 2. 

Hygiene and biosecurity routines on farms, in aquaculture and in feed production are important components of disease prevention. These routines were initially implemented as a spin-off effect from the eradication and control of specific diseases, such as bTB, bovine brucellosis, infectious bovine rhinotracheitis and enzootic bovine leucosis in the dairy sector; Aujeszky’s disease in the pig sector; and *Salmonella* in all food animal and feed production. These measures have, for example, eliminated the need for vaccination against the most commonly occurring viral infections in broiler production [44].

Implementing new scientific achievements and surveillance procedures has, in several cases, been essential for the introduction of organised health controls and eradication programmes. For example, the eradication of enzootic bovine leucosis (EBL) was facilitated by the introduction of a bulk milk test method [45], which to date, has been applied for several other viral infections and also parasitic diseases [46], and the use of the DIVA vaccine, which allows for the immunological differentiation of infected from vaccinated animals, facilitated the eradication of Aujeszky’s disease (AD) [47].

#### 3.2.2. Organised Health Services

In addition to the organised health controls described above, the industry continues to provide health counselling services, hereafter referred to as organised health services, which are fully financed by the industry. In the respective production sectors, both activities are carried out in an integrated way by the same veterinary organisations. The organised health services focus on infections and health disorders not covered by the organised health controls. They include annual visits to affiliated producers/farms by specialist veterinarians and visits upon call for actions to control herd health problems. In pig production, major improvements in health status have been achieved for traditionally dominant infections such as atrophic rhinitis [21], E. coli-related piglet diarrhoea [48] and swine dysentery [49], which have been eliminated as major problems. The same applies to Salmonella in all food animals, due to the zero-tolerance policy described above [21]. Considerable improvements in health status have also been achieved in the dairy sector [50] and the poultry sector [44,51].

The organised health services are also tasked with supporting the official surveillance of different infectious diseases [21] and other animal health projects, including on AMR, as described in annual reports to the Swedish Board of Agriculture [52]. 

#### 3.2.3. Limitation and Control of Import of and Trade in Live Animals

The import of live animals to Sweden was identified at an early stage as a risk factor for outbreaks of major epizootic diseases and other infectious diseases, and an early national policy for the food animal sector was, therefore, to limit the import of live animals [20]. This strategy was challenged when Sweden joined the EU in 1995. To minimise the risk of introducing diseases not covered by the harmonised rules for trade in cattle, pigs, sheep and their genes (semen and embryos) within the EU, the voluntary Swedish Farmers Disease Control Programme (SDS) was introduced by the industry in 1995 [53]. During recent years, the measures have been extended to prevent the introduction of methicillin-resistant *Staphylococcus aureus* (MRSA) in pig production [53]. Over more than 20 years of this control system, low-volume and high-biosecurity trade in live animals from abroad has been maintained [53]. In addition to SDS, corresponding voluntary measures by the Swedish Poultry Meat Association (SPMA) have been in place in the poultry sector since Sweden’s accession to the EU. 

The restricted import policy prevented the introduction of porcine reproductive and respiratory syndrome (PRRS) in pigs during its global spread in the late 1980s [54]. Likewise, the restricted and controlled trade in live pigs from abroad most likely mitigated the introduction of MRSA into pig production in Sweden [55]. In addition, the pre-import testing of day-old grandparent chickens in the late 1980s prevented the pandemic spread of S. Enteritidis from entering the Swedish poultry population, and has also considerably decreased the risk for the introduction of other serovars of Salmonella [32]. Voluntary efforts by SPMA, in cooperation with the National Veterinary Institute (SVA), have also mitigated the introduction of *Escherichia coli* resistant to extended-spectrum cephalosporins (ESC) in broiler production via breeding animals traded from abroad [56,57]. 

### 3.3. Summary of Key Factors

State veterinary leadership, veterinary infrastructure and regional veterinary laboratory capacity were established early in Sweden for the eradication of major epizootic diseases. These later became valuable tools in the prevention of other infectious diseases and facilitated the introduction in 1945 of industry-led health counselling services. The regulatory implementation of organised health controls in 1969 clarified the responsibility of the industry for disease prevention but also offered a tool for financial support from government for the control of specific diseases and other activities of national interest. This transformed the industry-led health counselling services into coordinated and focused activities. As a result, important endemic diseases were controlled or eradicated through joint action by government and the industry.

Access to veterinary expertise and regular visits by veterinarians to farms enrolled in organised health controls and organised health services provide farmers with farm-specific advice on the management and prevention of diseases and are also important for compliance with policies and recommendations on biosecurity, the use of antibiotics and good agricultural practice.

The limited and controlled import and trade in animals, genes and feed ingredients has prevented the introduction of several infectious diseases.

Hygiene and biosecurity routines on farms and in feed production were implemented early in Sweden through the control and eradication programmes. 

Although not specifically described here, or found to be specifically documented, stringent animal welfare regulations are also considered to have improved animal health and decreased the need for antibiotic treatment [58]. For example, limiting the stocking density in broiler and pig production [59], weaning after 26 days of age [60], prohibiting tail docking [61] and encouraging the use of straw bedding [61,62] in pig production are considered to have promoted health. These measures are supported by opinions from the European Food Safety Authority (EFSA) on the welfare of pigs [63,64].

## 4. Antibiotic Use and Resistance

### 4.1. Early and Continuous Awareness of the Problems with AMR

The concept of AMR and the need for the prudent use of antibiotics in animals were already familiar to Swedish veterinarians in the 1950s. Numerous articles in the national veterinary journal highlighted the issue and the risk of overuse [65,66,67,68,69,70]. In 1963, a session at the annual Swedish Veterinary Conference was dedicated to antibiotic use and AMR, with a Danish keynote speaker lecturing on the pros and cons of antibiotic use in animals [71]. In subsequent years, the issue was raised again in the national veterinary journal [72,73,74,75] and also in the trade journals [76,77,78,79,80]. At the annual Swedish Veterinary Conference in 1973, there was a public debate on the use of antibiotics in animals [81]. In that year, a Swedish study by Jonsson and Jacobsson [82] questioned the practice of providing in-feed medication to calves, and others presented AMR from a human healthcare perspective [83,84].

From the 1950s onwards, data on AMR in animal pathogens were widely presented (see below), making Swedish practitioners aware of AMR as a problem that could be encountered in everyday clinical practice (e.g., [85,86,87,88,89]). Further emphasis was placed on AMR as a concrete clinical problem following the documentation in the mid-1970s of transmissible resistance in *E. coli* from calves and pigs in Sweden [90,91]. Additional support for the discussion was provided by the first presentation of data on annual sales of antibiotics for animals (see below).

In the early antibiotic era, pharmacokinetic data on antibiotics were scarce and dosages were generally not based on hard scientific evidence. To improve the treatment of bovine mastitis, the distribution of penicillin to the udder was studied in Sweden as early as the 1950s [92]. Later, pharmacokinetic studies of several other antibiotics were performed [93,94,95,96,97,98]. Several of these studies questioned the dosing regimens applied, an issue specifically highlighted in a symposium held in 1981 [99,100,101,102]. In addition, therapeutic studies were performed as a basis for antibiotic treatment [103,104,105,106,107,108,109,110,111,112,113]. Since 1973, harmonised information (Fass Vet) on the characteristics, indications and dosages of pharmaceutical products, including antibiotics, licensed for use in animals has been compiled by the Swedish Association of the Pharmaceutical Industry (www.lif.se). Fass Vet is updated annually and distributed to all veterinarians in Sweden, electronically since 2017 (www.fass.se). The contents are approved by the competent authority (currently the Swedish Medicinal Products Agency).

### 4.2. Ban on AGPs

As a result of the national focus on the use of antibiotics and AMR described above, Sweden became the first country in the world to legislate on the withdrawal of antibiotics for growth promotion, and the use of AGPs was banned in 1986. At that time, AGPs were included in practically all feed for pigs and broilers in Sweden. The ban was preceded by an intensive debate in the media and the industry, and by scientific evaluations by national authorities and other stakeholders, as reviewed by Nordéus [114]. It was argued, for example, that society/consumers preferred animal production not to be dependent on the routine use of antibiotics. In response to this, the Federation of Swedish Farmers voluntarily issued a policy on the restrictive use of antibiotics in 1981 [114,115] and requested a total ban on AGPs in 1984, following which the total national ban on AGPs came into force in 1986 [114]. The implementation and consequences of the ban on AGPs and on the sales of antibiotics were reviewed by Wierup [116]. The ban also led to an increased focus on disease prevention by means other than the use of antibiotics [117].

When Sweden joined the EU in 1995, the national ban on AGPs was challenged, and Sweden was asked by the EU to provide scientific evidence for upholding the ban. To obtain relevant evidence, a thorough review of the pros and cons of AGP use was made [118]. The evidence provided and the recommendation by the EU Scientific Steering Committee to phase out the use of AGPs [119], combined with a lobbying effort from Swedish representatives to uphold the ban, was probably an important contributing factor in the subsequent complete ban on AGPs in the EU in 2006 [120].

### 4.3. Prevention and Control of Bacteria with Specific Resistance

Efforts have also been made to directly counteract the spread of bacteria with AMR of specific importance. For example, in 1995, a policy on the treatment of mastitis in dairy cows recommended that cows infected with penicillin-resistant *Staphylococcus aureus* should be culled instead of treated [121]. The reduction in the occurrence of penicillin-resistant *S. aureus* from 10% in 1985 [122] to about 1% in recent years [123] is probably largely an effect of adherence to this recommendation. In another example, an outbreak of tiamulin-resistant *Brachyspira hyodysenteriae* in pig herds in 2016 was actively curbed [124]. This organism causes swine dysentery in pigs, and tiamulin is vital for treatment in affected herds. Efforts to control the spread of *E. coli* resistant to ESC in broiler production and MRSA in pig production by the control of imported animals are other examples of work to contain specific types of resistance.

Legislative efforts are also in place to mitigate the spread of bacteria with specific resistance, with the detection of methicillin-resistant coagulase-positive *Staphylococci* (MRS) and carbapenemase-producing *Enterobacteriaceae* (CPE) in animals being made notifiable in 2008 and 2012, respectively [125]. Complementary legislation in 2013 [126] defined the management of MRS cases in cats, dogs and horses and made provisions for infection prevention control plans in veterinary practices. The legislation was accompanied by recommendations from the Swedish Veterinary Association (SVF) [127]. The impact of these measures is evident in the fact that CPE has not been detected in animals in Sweden and that the situation regarding MRS is favourable [128].

### 4.4. Access to Data on Use of Antibiotics

Data on veterinary sales of antibiotics for animals in Sweden have been compiled and analysed by the SVA since 1980 [129,130,131,132,133,134]. Earlier data from the human sector made it possible to compare patterns of antibiotic use in animals and humans [135]. Initially, the data were based on sales of antibiotics for animals from wholesalers to pharmacies, but from 2003, they were based on sales from pharmacies to animal owners (prescriptions) or to veterinarians (requisitions). Specific data on antibiotic use at the farm level are not available, and the identification of sales to specific animal species is limited but can be obtained by combining sales data and data from other sources, as outlined by Grundin et al. [136].

As of 2000, sales data have been reported and analysed by the SVA in the yearly reports from the resistance monitoring programme Svarm (see below). Since 2002, they have been released together with the corresponding data from the human sector (www.sva.se). In addition, sales data have been reported annually to the European monitoring system on antibiotic sales for animals since the start of the programme [10] and to the OIE since 2016 (OIE [137]).

Sales information is regularly communicated at conferences and meetings and also attracts general public interest. It is also used in the formulation of national guidelines on the use of antibiotics and in assessments of compliance with recommendations and regulations on the use and prescription of antibiotics.

### 4.5. Access to Data on Antibiotic Resistance

Data on AMR in animal bacteria have been compiled and presented since the 1950s. Initially, the data originated from prevalence studies and research projects (e.g., [85,86,88,90,138,139,140,141]. However, since 1978, resistance in *Salmonella* spp. from animals has been monitored yearly at the SVA and in 2000, this activity was extended to the Svarm programme, following a government decision. In Svarm, resistance in salmonella, campylobacter, indicator bacteria from healthy animals and several animal pathogens is monitored, and the data are presented and analysed in yearly reports. Since 2012, data from Svarm have been presented with corresponding data from the human sector compiled by the Public Health Agency of Sweden in an integrated report (Swedres-Svarm) available online (www.sva.se). 

Data on AMR from a clinical perspective are also regularly communicated to veterinarians at conferences and meetings, in general, highlighting the risks of the misuse and overuse of antibiotics. The data are also used in the elaboration of national guidelines on the use of antibiotics and for evaluating the effects of measures to mitigate resistance, for example, containing the spread of *E. coli* resistant to ESC in broiler production [57].

### 4.6. Policies, Guidelines, Recommendations and Legislation

Following the ban on AGPs, in 1990, the SVF issued guidelines on the use of antibiotics for the group treatment of pigs [142], and in 1999, it issued a general policy on the use of antibiotics in animals [143]. A policy on the treatment of mastitis in dairy cows was launched by independent experts in 1995 [121]. Later, the SVF gathered practitioners and experts on specific animal species and experts in the field of antibiotics to produce specific guidelines for companion animals, cattle, pigs, horses, sheep and goats. Those documents aim to balance the need for effective therapy with the need to minimise the emergence of AMR, for example, by advocating the use of narrow-spectrum antibiotics and avoidance of substances such as fluoroquinolones and third-generation cephalosporins, which are among the antibiotics of the highest priority of those categorised by the WHO as critically important in human healthcare [144]. The SVF guidelines are revised regularly and available on the website of the organisation (www.svf.se). To complement those policies and guidelines, since 201,2 the Swedish Medical Products Agency has issued detailed recommendations on the treatment of specific diseases in various animal species, which are available on the website of the agency (www.lakemedelsverket.se). 

Basic elements in the legislation since the 1950s on medicinal products for animals are that antibiotics may only be used in animals after prescription and that veterinarians may not sell antibiotics for profit [145]. Antibiotics may be prescribed to farm personnel for specified clinical conditions in farm animals without the prior examination of an animal by a veterinarian. This so-called “conditional use” is strictly regulated in legislation [146] and includes training courses for farmers and regular visits by the prescribing veterinarian. Since 2014, antibiotics of special importance in human healthcare, for example, glycopeptides, carbapenems and ceftaroline, are restricted from use in animals [146]. The use of fluoroquinolones and third-generation cephalosporins is limited by legislation to situations where laboratory tests show a lack of alternatives. To prevent the spread of contagious diseases at clinics and ambulatory practices, the legislation requires veterinary practitioners to have infection prevention and control plans [126]. These plans must also aim to mitigate the spread of resistant bacteria, for example, methicillin-resistant *Staphylococcus pseudintermedius* (MRSP), in animal healthcare.

### 4.7. Summary of Key Factors 

Since the mid-1950s, shortly after antibiotics became available for use in animals in Sweden, prudent use and the risks of AMR developing have frequently been discussed within the veterinary profession (practitioners and researchers). Antibiotics are considered important tools for the treatment of bacterial infections that need to be protected, not miracle drugs for dramatically improving animal production.

Data on the occurrence of AMR have been collected and disseminated in Sweden since an early stage and show the magnitude of the present and future risks for the veterinary and human sectors. Data on antibiotic sales, regularly reported since 1980, make it possible to assess compliance with legislation and policies. Access to data on AMR and on sales of antibiotics make it possible to devise policies, recommendations, guidelines and legislation, and also to evaluate the effect of actions taken. This has transformed general awareness of AMR into concrete knowledge on prudent use and into concrete actions to mitigate the emergence and spread of AMR. 

The ban on AGPs in 1986 decreased the sales of antibiotics substantially and put the focus on disease prevention by other means, including measures for improved animal management, feeding and housing. The discussions leading to the ban involved different actors, including farmers’ cooperatives, veterinarians, politicians and relevant authorities, with consumer confidence as an important argument. This made AMR, the use of antibiotics and sustainable animal production important issues on the political agenda, which is still the case [147].

## 5. Cooperation in Problem Solving

### 5.1. Control of Infectious Diseases

In the control of infectious diseases, there is generally constructive cooperation between authorities, industry and academia/veterinary expertise in Sweden. The major stakeholders are the government; the competent authorities; veterinary and other academics; industry, representing farmers, slaughterhouses and dairies; and also animal breeding and animal feed companies. The Federation of Swedish Farmers acts as a link between the government and industry. Examples of formal cooperation are the steering committees for the organised health controls, which cooperate with national authorities and industry. However, “ad hoc groups” have also been formed for other animal health, food safety and public health issues, such as *Campylobacter*, *Salmonella* and enterohemorrhagic *Escherichia coli* (EHEC), and for managing the use of zinc oxide in pig production. A large programme introduced in 2016, with funding from the Swedish Board of Agriculture, focuses on biosecurity on individual holdings, aiming at reducing the spread of infectious diseases [148]. The programme is run by the industry-owned animal health service providers.

For the control of endemic diseases, including associated interventions on individual animal holdings, the industry-led veterinary organisations, and experts from academia and from the SVA are the most significant players. Clinical practitioners also play an important role and, together with farmers, are the first to observe and highlight animal health problems, including those related to AMR. Major animal health-related control measures are initially devised by academia, the SVA or industry, and formally elaborated and developed further at meetings chaired by the competent authority.

A One Health perspective has been used to address zoonotic infectious diseases since the Zoonosis Council, including representatives from the human sector, was established as a national collaborative forum for authorities and organisations in 1997 [149]. At council meetings, strategies in the zoonosis area are elaborated and discussed, to achieve mutual understanding between all the authorities and organisations involved.

### 5.2. Counteracting AMR

There is also cooperation between stakeholders on specifically addressing AMR, in a way similar to that described for infectious diseases above. For example, since 2005, the SVA and Farm and Animal Health, the industry-owned animal health services provider, have cooperated on monitoring AMR in food animals within the SvarmPat project [128]. Likewise, the SVA cooperates with the SPMA on issues related to AMR in broiler production, for example, on reducing the occurrence of ESBL [128]. Moreover, recommendations and guidelines on the use of antibiotics and measures to mitigate the spread of AMR are elaborated with the participation of experts from several sectors (see above). In 2008, the network Strama VL was started as a platform for enabling stakeholders in the veterinary sector to exchange information, analyse problems, pinpoint solutions and initiate prioritised activities. A secretariat at the SVA coordinates the network and acts as a contact point and a centre of knowledge [150].

A One Health perspective on AMR has been successively introduced by increased cooperation between the veterinary and human sectors. Thus, when the Swedish strategic programme against antibiotic resistance (Strama) was formed in 1995 to counteract AMR in the human sector, the veterinary sector was involved in the network [151]. The cooperation against AMR between sectors was eventually formalised in 2012 by the creation of the Intersectoral Coordinating Mechanism (ICM) chaired by the Public Health Agency of Sweden and the Swedish Board of Agriculture [152]. The ICM brings together representatives from about 20 national authorities and organisations across many sectors: human and animal health, food production, the environment, research, trade and international relations. In the early 2000s, the National Board of Health and Welfare, in cooperation with stakeholders from several sectors, elaborated a proposal for a national strategy on AMR, which was presented to the Swedish Parliament in 2005 [153]. Moreover, under the auspices of the ICM, action plans on AMR for authorities were presented in 2015 [154] and revised in 2017 [155]. Other examples of cooperation are the presentation and analysis of data on antibiotic sales and AMR by the National Veterinary Institute and the Public Health Agency of Sweden in yearly Swedres-Svarm reports (see above) and various research activities, for example, the IMPACT project [156].

### 5.3. Agricultural Policy with Incentives for Livestock Production 

Until the early 1990s, Swedish agricultural policy included incentives for livestock production. Apart from broiler production and some minor sectors, domestic food production was supported by a differentiated system of price regulation and other budget measures. This facilitated government support of the animal health sector until Sweden’s accession to the EU in 1995. Before then, the Federation of Swedish Farmers held regular formal meetings and negotiations with the government, which included discussions on investments in the animal health sector. This pattern of cooperation has been maintained, although economic accountability has been modified, and economic support from the government has generally decreased substantially in recent years. 

The costs for the control and eradication of the major epizootic diseases (Table 1) have, to date, been covered by the Swedish government. In the case of salmonella, farmers’ costs for eradication were fully covered until 1984, when the compensation was reduced for cattle and pig farms and withdrawn for broiler production [20]. Today, compensation is higher for herds affiliated with organised health controls focusing on biosecurity.

Government support facilitated the creation of the industry-led health counselling veterinary organisations for the prevention of endemic infection. In 1969, the industry became financially responsible for that activity, but up to Sweden’s EU accession in 1995, the government continued to provide financial support for the control and eradication of specific diseases through organised health controls (see above, Table 2). That financial support was used to create economic incentives for producers to join control programmes. In addition, animal health insurance companies require farms to have disease prevention measures in place, for example, an affiliation with an organised health control, to qualify for compensation for outbreaks of a disease. Today, the financial support from government is mainly limited to funding the control organisations to perform specific activities considered of national importance, for example, disease surveillance. In addition, since 1934, the government has provided financial support to the district veterinary organisations to ensure that clinical animal health services are available to farmers in the whole of Sweden [20].

### 5.4. Summary of Key Factors

Using regulatory and financial tools, the competent authority in Sweden has facilitated active control solutions for infectious diseases and AMR. 

Cooperation between relevant stakeholders has enabled mutual understanding of and consensus on the need for and benefits of implementing measures to prevent and control infectious diseases and to counteract AMR. 

Cooperation between national authorities in the human and animal sectors from a One Health perspective has strengthened the basis for strategies against zoonotic diseases and AMR in the animal sector. Research cooperation between the two sectors has further increased consensus.

## 6. Discussion

In Sweden, sales of antibiotics for use in animals and the prevalence of AMR in food animals are low in comparison with those in other EU Member States [11,12]. Moreover, in 2019, more than 90% of the overall sales of antibiotics for use in animals consisted of products for the treatment of individual animals, 58% of which were narrow-spectrum benzylpenicillin [128]. These data indicate a limited need for antibiotic therapy and adherence to the principles of prudent use set out by the European Medicines Agency (EMA) [157]. This case study shows that the favourable situation in Sweden is the result of several factors.

### 6.1. Early Awareness

A factor of basic importance is probably the early and now-widespread awareness among veterinarians and farmers of the risks of AMR and the need for the prudent use of antibiotics. This is likely due to the discussions on antibiotic use in animals starting early in Sweden, in the 1950s. These discussions were supported by access to data on antibiotic use after 1980 and early detailed national reports on the occurrence of AMR in bacteria from animals. The Swedish ban on AGPs in 1986, the first in the world, was actually requested by farmers, which exemplifies the general awareness of AMR even at that time. The request for a ban was probably also a response to the ongoing public debate on the state of animal husbandry that started in the mid-1970s and threatened to undermine consumer trust in Swedish food animal production [114,115]. 

### 6.2. Cooperation in Prevention and Control of Infectious Diseases

A major reason for the low current use of antibiotics is that the need for antibiotic therapy has been reduced by successful long-term efforts to prevent and control infectious diseases and to eradicate these diseases when possible. These efforts started in the pre-antibiotic era and demonstrate a committed attitude to the prevention of infectious diseases. In outbreak situations, substantial efforts are made to trace and eliminate the source of infection, making it possible for Sweden to achieve disease-free status even for diseases such as bovine paratuberculosis [21,158] and effectively control and eliminate *Salmonella* from infected herds and feed mills [21]. An even more important factor in reducing the need for antibiotics is that similar strategies are applied for endemic diseases, applying a concept of “prevention is better than cure”, as discussed below.

A significant factor for the achievements in animal health in Sweden is the longstanding cooperation between relevant stakeholders. This enables mutual understanding of and consensus on the need for and benefits of controlling diseases, even among stakeholders with different economic priorities, usually at meetings with a national perspective chaired by the competent authority with its regulatory and economic power, including both a “carrot and stick”. For example, active participation by farmers’ organisations has been identified as a strong factor in the control of bovine viral diarrhoea (BVDV) [159]. Consensus was reached as early as the 1950s on regulations treating animal feed as a potential source of infection, i.e., a “feed to fork” perspective, in contrast to the “farm to fork” perspective still applied in the EU [160]. Measures to mitigate AMR have been decided in line with the so-called “Swedish model”, also accounting for consumer opinion [16,136]. Cooperation and consensus between stakeholders has also been identified as a key factor in the success of work against AMR in Danish pig production [161].

### 6.3. Organised Health Controls for Prevention of Endemic Diseases

The introduction of organised health controls in 1969 was strategically important, because it opened the door for indirect government leadership in the control of endemic diseases, in cooperation with industry and its veterinary organisations. In other countries, government and regulatory influence in the prevention and control of endemic diseases is often lacking, although these diseases cause the highest burden in animal production and are the major targets for antibiotic treatments [162]. Furthermore, Sweden considers the control and eradication of viral diseases, which predispose for bacterial infections that require antibiotic treatment (e.g., BVDV, EBL, MV, PRRS, AD, Caprine arthritis encephalitis and infectious bursal disease) to have health-supporting effects beyond their direct clinical impact [117,163]. In addition to the eradication of several infections (Table 2), the organised health controls cooperate in the surveillance of infectious diseases and were, for example, early in identifying the introduction of PRRS in 2007, allowing it to be successfully eradicated [164,165]. In cases of health problems on individual farms, the ambition is to identify and eliminate the basic problem, i.e., “problem solving at the root”, instead of alleviating clinical problems by using antibiotics.

The long-term disease preventive measures taken in collaboration between government agencies and other stakeholders have improved animal health and decreased the need for the use of antibiotics. The best example of this is probably Swedish commercial broilers, with a yearly production of about 100 million chickens, where only 0.25% of flocks (8 of 3178) were treated with antibiotics in 2018 [166].

### 6.4. Financial Incentives

The national incentives for livestock producers, initially introduced in Sweden to ensure national food security in the event of war and to facilitate the structural rationalisation of the farm industry, limited the cost to individual farmers of joining disease control programmes until Sweden’s entry into the EU in 1995. In addition, the price regulation system during that period and limited imports of animal-derived food products minimised the risk of producers losing market share due to the costs of disease control. This facilitated the implementation of costly eradication programmes, which were funded on a national basis owing to strong arguments, primarily from veterinarians and industry. This situation may reflect a Swedish attitude to respecting facts and trust among farmers in politicians and relevant animal health authorities, including the industry’s own veterinary organisations. Although the control and eradication of diseases in farm animals may seem to be associated with high costs, the actual burden of disease justifies the costs of measures for disease control and hygiene [167]. This is exemplified by the eradication of AD in pigs [168], BVDV in cattle [169] and swine dysentery in pigs [170]. Additionally, when avoided costs for human illness are considered, it has been shown that for salmonella, the benefits exceed the costs for the control [171,172].

### 6.5. Access to Data on Sales of Antibiotics and AMR

Data on sales of antibiotics and the occurrence of AMR have made the discussion on the use of antibiotics more concrete but have also formed the basis for measures and actions and for the evaluation of actions taken. Recommendations and policies on the therapeutic use of antibiotics and infection control procedures have been regularly updated, and actions against resistance of specific importance, for example, MRSA, have been developed, including legislative measures. Policies and guidelines are generally well received and valued by Swedish veterinary practitioners [173,174] and also by Swedish farmers [175]. This approach to interventions against AMR in the face of unwanted trends and new knowledge was highlighted by the Directorate-General for Health and Food Safety (DG-SANTE) of the European Commission as an important factor for the favourable situation in Sweden [176], and is in line with the action plans against AMR issued by the WHO [1] and the European Commission [13].

### 6.6. Early Action

It is interesting to note that in three areas of major importance for counteracting AMR, Sweden took control actions long before other countries [136]. The control and eradication of several diseases, for example, *Salmonella* infections, bTB and brucellosis, was initiated early. The successful control of *Salmonella* has been recognised internationally; for example, in 1993, Sweden was engaged by the WHO to teach others about *Salmonella* control in poultry [177]. In 1980, Sweden became the first country in the world to publish data on the sales of antibiotics [133], and in 1986, it became the first country to ban the use of AGPs [114]. The experiences gained in Sweden were valuable when a ban on AGPs was introduced later in other countries, for example, the other Nordic countries [178,179]. Sweden was also comparatively early in setting up a comprehensive national monitoring programme for AMR in animals (2000). Furthermore, Sweden was the first country in the world (1986) to ban the use of meat and bone meal from fallen stock or sick animals in animal feed, which apparently protected Sweden from an outbreak of C-type Bovine spongiform encephalopathy BSE [180,181].

### 6.7. Geographical Location

This review did not find any evidence that the geographical location of Sweden or differences in, for example, climate or the intensity and productivity of food animal production could explain the favourable animal health situation and, thereby, the low use of antibiotics. The climate in the regions of Sweden where most food animal production is located is similar to that in northern continental Europe, and the spread of vector-borne diseases to new regions attributed to global warming, for example, bluetongue [182] and Schmallenberg virus [183], has also affected Sweden. Sweden, like several other EU Member States, has a long sea border, which decreases the risk of the uncontrolled transboundary movement of animals and animal products. However, in most countries, outbreaks of transboundary diseases in the past were mainly caused by the regular trade and imports of live animals, before these were properly regulated [184]. We found no indications that the conditions for the dissemination of infectious diseases are different in Sweden than in other developed countries. Before effective controls were in place, transboundary diseases introduced into Sweden (Table 1) were spread widely, often up to the far north of the country above the polar circle.

Food animal production in Sweden has, in principle, undergone the same structural changes as in most other developed countries. The spread of infections between and within farms is mainly limited by the biosecurity measures applied on individual farms, which may be more difficult to achieve in countries with denser animal populations than Sweden. However, the size of the national food animal production sector does not appear to be a factor explaining the favourable situation regarding AMR in Sweden [136,161]. For example, the consequences of the withdrawal of AGPs in Danish pig production were very similar to those observed in Sweden 20 years earlier, despite the much larger pig production sector in Denmark [116,185]. Since the productivity in Swedish animal food production is generally on the same level as in other developed countries, it can be concluded that it is possible to combine high productivity in animal production with a restricted use of antibiotics, as also found, for example, in Denmark [161]. The experiences from Sweden are, therefore, generally applicable to other countries.

### 6.8. Future Challenges

A major future challenge for Sweden is to find ways to maintain the good status achieved for animal health and AMR while not contravening harmonised EU rules on the intracommunity movement of animals and international standards on trade in live animals. For example, it is of vital importance for Swedish pig production to prevent the introduction of PRRS, which, apart from causing economic losses, would be a major trigger for the increased use of antimicrobials, as found in other countries [163]. In order to comply with the WHO [1] and EU [13], in action plans against AMR, it is therefore necessary for Sweden to find transparent and evidence-based procedures for preventing the introduction or reintroduction of important infections and bacterial strains with special antimicrobial resistance. The new EU Animal Health Law [186] and Veterinary Medicines Regulation [187] provide significant incentives for individual countries and farmers to maintain or improve their animal health status, thereby decreasing the need for antimicrobials, and to strive for a decreased and prudent use of antimicrobials, thereby lowering the risk of the emergence of AMR. These are all necessary steps for EU-wide and global progress in efforts to contain AMR. Another challenge for Swedish pig production is to cope with the phasing out of zinc oxide in the EU [188] without increasing antibiotic use. The ongoing trend towards larger pig and dairy herds might also facilitate the spread of infectious diseases and thereby pose a challenge for infection control. An overall challenge for Swedish producers is to maintain ambition and economic power for animal health investments, which from a short-term perspective, may not be rewarded on the open market. It is crucial to maintain understanding/conviction regarding the overall benefits of healthy animals.

## 7. Materials and Methods

We retrieved and analysed documents published from Sweden since the early 1900s with the aim of identifying factors and measures undertaken that are likely to have contributed to the favourable situation regarding antibiotic use and AMR. The work was structured around the three major areas: 1. the prevention and control of infectious diseases; 2. antibiotic use and resistance; 3. cooperation in problem solving. The focus was on farm animals, and as a background, data on the food animal production during the same period were retrieved.

To retrieve scientific documents, we used PubMed searches, and to retrieve grey literature and legislative documents, Google searches were used. In addition, all issues of the journal of the Swedish Veterinary Association since 1945 and the programmes of the yearly national veterinary conferences since the mid-1950s were scrutinized for articles and other material related to antibiotics. Official inquirers related to the animal sector and policy documents on the use of antibiotics were also consulted. In addition, we received valuable information and advice from specific key persons (see Acknowledgements) with insight into the animal health sector and the work against AMR in Sweden. These key persons also read and commented on the first draft of this manuscript.

## 8. Conclusions

This case study revealed that a fundamental factor in Sweden’s favourable status as regards antibiotic sales and AMR was an early insight into and sustained awareness of the risks of AMR and the need for prudent use to maintain the efficacy of antibiotics. Early access to annual data on antibiotic sales and AMR provided insights and made it possible to focus measures on areas of concern and to assess the impact of these measures.

Another major factor has been the long-term control and eradication of infectious diseases, including endemic diseases, which reduced the need for the use of antibiotics. The structures and strategies established for that purpose, under government leadership and support, also proved useful for counteracting AMR as an integral part of disease prevention and control.

A third vital factor for success has been the consensus among relevant stakeholders on the need to address AMR and control infectious diseases, and stakeholder cooperation in the design and implementation of relevant measures.

In summary, early awareness, longstanding efforts to prevent diseases and consensus between stakeholders explain the success of Swedish work in preventing the development of AMR.

## Figures and Tables

**Figure 1 antibiotics-10-00129-f001:**
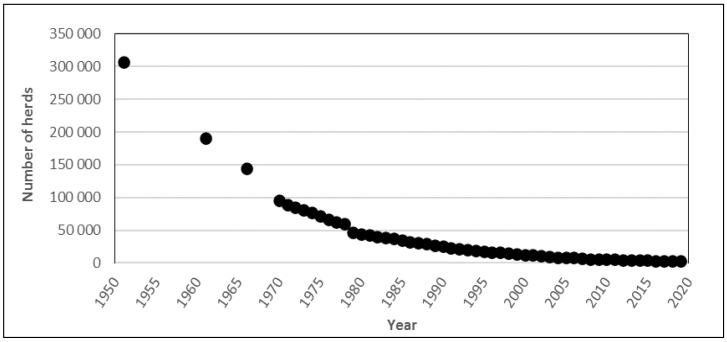
Numbers of cattle herds (1951–1978) and dairy herds (1979–2018) in Sweden. Data from the Swedish Board of Agriculture [14]. The data for 1951–1978 include all types of cattle herds, but the majority of farms in that period can be assumed to have had milking cows, i.e., to have been dairy herds.

**Figure 2 antibiotics-10-00129-f002:**
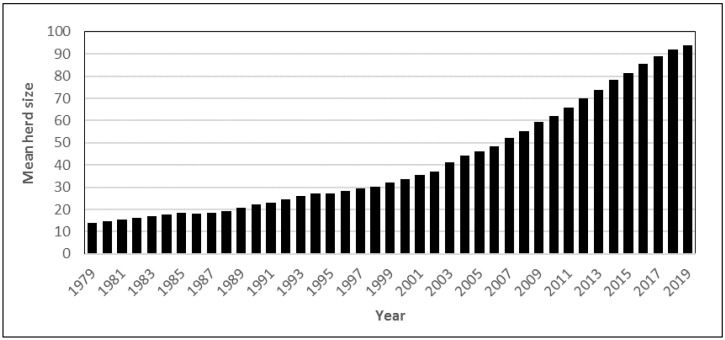
Mean dairy herd size for 1979–2018 in Sweden. Data from the Swedish Board of Agriculture [14].

**Figure 3 antibiotics-10-00129-f003:**
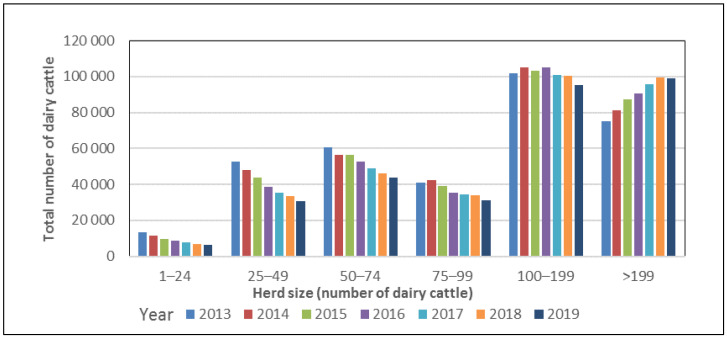
Number of dairy cattle in different herd size categories for 2013–2018 in Sweden. Data from the Swedish Board of Agriculture [14].

**Table 1 antibiotics-10-00129-t001:** Significant outbreaks of major epizootic diseases in Sweden for 1900–2020. Adapted from Cerenius [20], National Veterinary Institute (SVA) [21] and [22].

Year	Disease	Animal Species	Comments
1924–27	Foot and mouth disease (FMD)	Cattle	11,002 herds infected.
1938–40	FMD	Cattle	7293 herds infected.
1940	Classical swine fever (CSF)	Pigs	230 herds infected.
1943–44	CSF	Pigs	445 herds infected.
1950–56	Paratuberculosis	Cattle	Beef cattle, 830 animals seropositive.
1951–52	FMD	Cattle	562 herds, 1 million cattle vaccinated.
1953	Salmonella epidemic	Several, mainly cattle	9000 human cases, 90 deaths.
1956–57	Porcine brucellosis	Pigs	76 herds infected.
1956–57	Anthrax	Cattle/pigs	19 cattle herds/68 pig herds infected.
1960	FMD	Cattle	6 herds infected.
1993	Paratuberculosis	Cattle	53 beef cattle herds infected.
1991–97	Bovine tuberculosis	Farmed deer	13 herds infected.
1995–96	Newcastle disease (ND)	Poultry	650 flocks tested; 1.75 million birds/eggs destroyed.
2007	Porcine reproductive and respiratory syndrome (PRRS)	Pigs	7 herds infected, modified stamping out.
2008–09	Bluetongue	Cattle	30 outbreaks in different regions, 2.7 million cattle vaccinated.
2010–20	Highly pathogenic avian influenza and ND	Poultry	2 and 5 outbreaks, respectively.
2010–20	Anthrax	Cattle	12 outbreaks.

**Table 2 antibiotics-10-00129-t002:** Examples of infectious diseases eradicated (recognized free by EU) or brought under control by organised health controls in Sweden [21,43].

Year	Disease	Animal Species	Comment
1990–2001	Enzootic bovine leucosis (EBL)	Cattle	Initially, 25% of dairy herds infected. Eradicated.
1991–1996	Aujeszky’s disease (AD)	Pigs	Initially, 5% of sow herds infected. Eradicated.
1993–2013	Bovine viral diarrhoea (BVDV)	Cattle	Initially, 40% herd prevalence. Nationally declared freedom in 2014.
1993–	Maedi-Visna (MV)	Sheep	Initially, 8.2% herd prevalence. Ensure disease-free herds for livestock trade.
1994–1998	Infectious bovine rhinotracheitis (IBR)	Cattle	Initially, 0.2% of dairy herds seropositive. Eradicated.
1998–2017	Paratuberculosis	Cattle	Repeated introduction in beef cattle after imports; latest case in 2005. Dairy herds never infected. Surveillance to document a very low prevalence or possibly declare national freedom.
1998–	Porcine reproductive and respiratory syndrome (PRRS)	Pigs	One outbreak in 2007. Freedom nationally redeclared in 2008.
1993–	Campylobacter spp.	Broilers	Surveillance aiming at reducing prevalence.

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
