# Peer review of "Successful Prevention of Antimicrobial Resistance in Animals—A Retrospective Country Case Study of Sweden"

_antibiotics, 2021, doi:10.3390/antibiotics10020129_

Round 1

Reviewer 1 Report

This is very well written paper exploring the reasons behind Sweden’s relative success in keeping the use of antibiotics low in animals (as well as a relatively low level of antibiotic resistance overall). The paper provides a useful and interesting history of ID outbreaks in Sweden over the last century. It also gives a particularly interesting description of governance arrangements and the authority of the state versus industry. Also provides a very nice overview of awareness and the rise in questioning of antibiotic use in recent decades.

Clearly other countries have much to learn from the Swedish experience – especially with regards to cooperation and the use of seemingly appropriate incentives.

Minor comments

The Discussion seems to repeat several of the points from the results. I wonder if perhaps sectioning it with several clear subtitles might help distinguish it..(?)

Would be useful to define cattle herd versus dairy herd as not all readers will have a agriculture/animal background

Why the increase in yield per cow?

Line 87 – shouldn’t it say, “BUT, for commercial broiler production, the number of farms has increased.” ?

Line 91 – “FROM an international perspective.”

Line 100 sounds more normal to say “in the same EFFICIENT (or EFFECTIVE) way…”

Line 128 – “is notifiable” – do you mean “must be reported”?

Line 136 – could you define “layers”

Line 143 – should be “other ingredients at risk…”

“The introduction of more industrialised food animal production processes increased 159 health disorders caused by endemic diseases, mainly respiratory infections.” Ref?

Line 168 “A basic task for the health control in pig production” (no need for “the”)

Should define “outbreak”

Should define “certified freedom”

Possible to say anything about space provided per animal for healthy conditions?

Line 238 – possible to describe the safe trade policy?

Reviewer 2 Report

The manuscript presents a detailed analysis of policy interventions to control AMR and antibiotic use in food producing animals in Sweden. Overall, it is well researched and written and would be of great interest to the readers of antibiotics. However, it is very long in its current state and is at times repetitive in the discussion from the results. In addition, some of the results need to be streamlined by focusing on one or two examples that present the point the authors wish to make rather than giving such detail and length.

General comments

The authors use ‘data’ throughout as singular. This should be plural and should be corrected throughout the manuscript.

Food animal production – section 2

The authors address the three production sectors of dairy, pigs and broiler production. What is the justification for choosing these sectors? Are these the largest sectors by production capacity in Sweden? The section would benefit from an introduction to the species covered and the justifications for focusing on these sectors.

The authors address herd size and number of farm holdings but do not explore any further detail on changes to management practices. Have the increase in herd sizes been accompanied by intensification of production? Have management practices changed within the time periods discussed? This would be a welcome addition to this section.

Prevention and control of infectious diseases – section 3

Overall, this section is too long and includes too much specific detail. It would be beneficial for the length to be reduced and the authors to focus on one or two examples in detail. For example, salmonella.

The authors introduce that salmonella control is novel and different in Sweden to other countries (lines 119-121). It would be beneficial to include a brief introduction about what makes the measures taken in Sweden different to other countries.

Antibiotic use and resistance – section 4

Section 4.4 (lines 366-377) outline the veterinary antibiotic sales data collected in Sweden. Is there any monitoring of farm level use of antibiotics? If so, this would be a welcome addition to describe in this section.

The authors needs to take care not to use the terminology of ‘use’ in this section if it is referring to antibiotic sales data. Use on farm is a different metric to what antibiotics are sold.

Discussion

The discussion is quite repetitive from the results. The authors need to focus on the comparative aspects of the discussion and remove some of the discussion which simply repeats the previous commentary. For example, (lines 546-556) is repeating what is stated in the results and does not really address discussion of the results.

Materials and Methods

What specific methodological approaches were used to policy analysis? This would benefit from more specific detail on the approaches used.

Specific comments

Lines 27-28 – The development of resistance is a natural phenomenon. However, the overuse of antibiotics can accelerate its development. Please re-phrase this sentence so that this is clear.

Line 142 – There is an additional space after ‘revealed’ that needs removing.

Line 159 – What is meant by ‘industrialised food animal production’? Does this mean more intensive production? For example, a move towards zero grazing systems in dairy cattle? Higher stocking densities in broiler and pig production? Please add further details to justify this point.

Line 175 – ‘196os’ should have a zero and be ‘1960s’ and not the letter ‘o’.

Line 187 - There is an additional space after ‘[41],’ that needs removing.

Line 227 – What is the DIVA vaccine? This is presumed to be a vaccination against Aujeszky’s disease but this is confusing terminology. Ensure this is clear.

Line 283 – Define the abbreviation ‘EFSA’.

Lines 401-402 – This would benefit from inclusion of a line on the WHO highest priority critically important antibiotics to justify the importance of cephalosporins and fluoroquinolones.

Line 452 – Define the abbreviation ‘EHEC’.

Line 529 – Missing a full stop at the end of the sentence.

Line 545 – Define the abbreviation ‘EMA’.

Line 555 - ‘197os’ should have a zero and be ‘1970s’ and not the letter ‘o’.

Line 620 – What is the ‘DG-SANTE’? This is not clear and requires further clarification.

Author Response

Reply to reviewers submitted in separate file

Round 2

Reviewer 2 Report

Many thanks for your extensive response and amendments. Whilst I appreciate that the manuscript is a detailed review I still feel that it would be more manageable to the reader if its overall length was reduced and the content was streamlined now. However, as this concern was not mirrored by the other reviewer or editor I am willing to recommend acceptance of the manuscript in its current form. I do appreciate the authors have made significant efforts to respond to my previous comments.